# Multicenter Morphometric Analysis of Stratum Corneum Nanotexture for Skin Barrier Assessment

**Jen-Hung Wang**[1,2] (iD)                                      JEN-HUNG.WANG@SCILIFELAB.SE

**Chia-Yu Chu**[3] (iD)                                                  CHIAYU@NTU.EDU.TW

**Felipe Colombelli**[1] (iD)                                           FELCO@SCILIFELAB.SE

**Ching-Wen Du**[3] (iD)                                              JUDYDU@NTUH.GOV.TW

**Maria Oberländer Christensen**[4] (iD)     MARIA.OBERLAENDER.CHRISTENSEN@REGIONH.DK

**Jorge Pereda**[2] (iD)                                           JORGEPEREDA@GMAIL.COM

**Ivone Jakasa**[5] (iD)                                                    IJAKASA@PBF.HR

**Sanja Kezic**[6] (iD)                                          S.KEZIC@AMSTERDAMUMC.NL

**Jacob P. Thyssen**[4] (iD)               JACOB.PONTOPPIDAN.THYSSEN@REGIONH.DK

**Edwin En-Te Hwu**[*2] (iD)                                                ETEHW@DTU.DK

**Gisele Miranda**[*1] (iD)                                   GISELE.MIRANDA@SCILIFELAB.SE

[1] *Science for Life Laboratory, Department of Computational Science and Technology, KTH Royal University of Technology, Stockholm, Sweden*

[2] *Department of Health Technology, Technical University of Denmark, Kongens Lyngby, Denmark*

[3] *Department of Dermatology, National Taiwan University Hospital and National Taiwan University College of Medicine, Taipei, Taiwan*

[4] *Department of Dermatology, Bispebjerg and Frederiksberg Hospital (BFH), University Hospitals of Copenhagen, Copenhagen, Denmark*

[5] *Laboratory for Analytical Chemistry, Department of Chemistry and Biochemistry, Faculty of Food Technology and Biotechnology, University of Zagreb, Zagreb, Croatia*

[6] *Department of Public and Occupational Health, Amsterdam Public Health Research Institute, Amsterdam University Medical Center, Amsterdam, The Netherlands*

**Editors:** Accepted for publication at MIDL 2026

## Abstract

Stratum corneum nanotexture (SCN) has emerged as a promising non-invasive biomarker for quantifying skin barrier impairment and the severity of inflammatory skin diseases such as atopic dermatitis (AD). In this multicenter study, we analyzed stratum corneum tape-strip samples from 90 patients with AD and 30 healthy controls recruited in Taiwan and Denmark, yielding a heterogeneous dataset of more than 2,000 SCN images. Participants were evenly stratified into four AD severity groups defined by the Eczema Area and Severity Index (EASI), enabling robust evaluation of SCN-derived metrics across the full spectrum of disease severity. Previous studies have primarily relied on count-based measures to quantify the density of circular nano-size objects (CNOs) in SCN images from single-center cohorts, without leveraging instance-level segmentation or comprehensive morphometric profiling. In this study, we propose and validate a segmentation-based SCN analysis pipeline that integrates YOLOv12 with Segment Anything Model 3 (SAM3) for accurate CNO delineation in a multicenter setting. This framework enables the extraction of detailed morphometric descriptors and facilitates systematic evaluation of SCN-derived biomarkers for quantitative skin barrier assessment in AD. Our code is available at https://github.com/mirandaresearchlab/SCN-SAM.

---

[*] Contributed equally

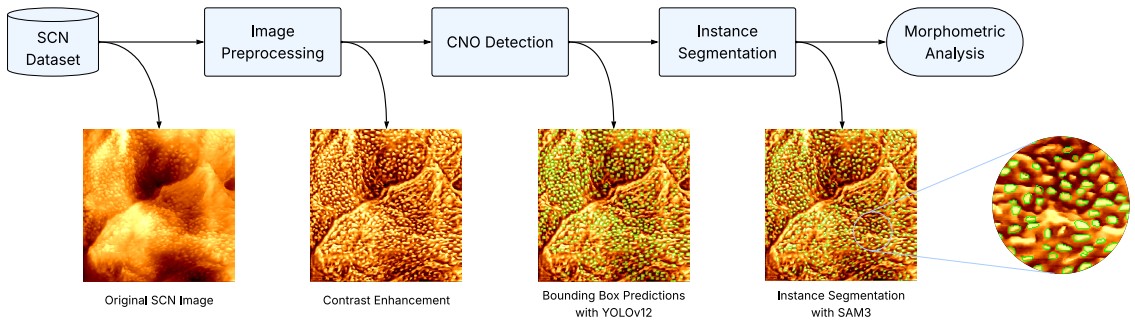

Figure 1: Overview of the proposed segmentation-based morphometric analysis pipeline for stratum corneum nanotexture (SCN).

**Keywords:** Atopic Dermatitis, Object Detection, Instance Segmentation, Morphometric Analysis, Stratum Corneum Nanotexture

## 1. Introduction

Atopic dermatitis (AD) is a chronic, relapsing inflammatory skin disease characterized by intense pruritus, eczematous lesions, and a substantial impact on quality of life (Langan et al., 2020). Globally, AD is estimated to affect approximately 790 million individuals, including about 20% of children and 10% of adults (Silverberg et al., 2021). In clinical practice, disease severity is most commonly assessed using the Eczema Area and Severity Index (EASI), which combines the extent of body surface involvement with the intensity scores of key clinical signs (i.e., erythema, edema/papulation, excoriation, and lichenification) across four body regions (Hanifin et al., 2001). Despite its widespread use, EASI relies on subjective visual assessment and demonstrates only moderate interrater reliability (Schmitt et al., 2013). Consequently, there is growing interest in objective, quantitative biomarkers that can capture epidermal barrier function and disease severity in a reproducible and scalable manner.

In recent years, stratum corneum nanotexture (SCN), the nanoscale topography of corneocyte surfaces, has emerged as a promising non-invasive biomarker of skin barrier function and disease activity, with potential to assess inflammatory skin disorders such as AD (Thyssen et al., 2020). SCN can be measured from stratum corneum (SC) tape strips (Clausen et al., 2016), a minimally invasive sampling method that can be combined with high-speed dermal atomic force microscopy (HS-DAFM) (Liao et al., 2022) to obtain high-resolution images of corneocyte surface architecture and its nanoscale alterations (Pereda et al., 2024). In particular, the presence and distribution of circular nano-size objects (CNOs) on the corneocyte surface have been associated with impaired barrier function and filaggrin-related abnormalities, indicating that SCN-derived metrics could serve as quantitative biomarkers of disease severity in AD (Franz et al., 2015).

However, prior SCN studies have mainly relied on simple count-based measures of CNO density, without leveraging instance-level segmentation to capture the detailed morphol-

ogy of individual CNOs (Riethmuller et al., 2015; Wang et al., 2024). When segmentation has been attempted, approaches have typically employed heuristic thresholding or hand-crafted image-processing pipelines, which are vulnerable to noise, artifacts, and variations in imaging conditions (Riethmüller, 2018). Moreover, most existing methods were developed and evaluated on single-center cohorts, which limits their generalizability and does not capture the heterogeneity of clinical practice, imaging protocols, and patient populations (Engebretsen et al., 2018a,b).

In this study, we validate SCN-derived metrics in a multicenter AD cohort comprising 90 patients with AD and 30 healthy controls recruited in Taiwan and Denmark, yielding a heterogeneous dataset of more than 2,000 SCN images that covers the full spectrum of disease severity. Building on recent advances in deep learning, we adopt a two-stage pipeline in which a YOLOv12 (Tian et al., 2025) model generates CNO detection bounding boxes as prompts for the Segment Anything Model 3 (SAM3) (Carion et al., 2025), enabling accurate instance-level segmentation and subsequent morphometric profiling of individual CNOs. To evaluate the clinical utility of the proposed segmentation-based SCN morphometrics, we assess their robustness across centers and Fitzpatrick skin phototypes (Fitzpatrick, 1988), as well as their ability to capture skin barrier impairment and AD severity.

## 2. Method

### 2.1. Study cohort and sample collection

This multicenter study enrolled 120 adults (age $\geq$ 18 years), comprising 90 patients with AD and 30 healthy controls, recruited in equal numbers from National Taiwan University Hospital (Taipei, Taiwan) and Bispebjerg and Frederiksberg Hospital (Copenhagen, Denmark). The cohorts have been described previously in studies with different objectives (Wang et al., 2024; Du et al., 2025). Participants were evenly divided into four groups (n=30 per group) based on AD history and EASI scores: healthy controls (no AD), mild AD (0 < EASI $\leq$ 7.0), moderate AD (7.0 < EASI $\leq$ 21.0), and severe AD (EASI > 21.0). Patients with other chronic skin diseases, active skin infections, or recent application of topical corticosteroids to the sampling sites (within the preceding three days) were excluded. No specific instructions were given to discontinue topical treatments, ensuring that the collected SC samples reflected routine clinical practice.

For each participant, SC samples were collected using a standardized tape-stripping procedure (Dapic et al., 2013) with circular adhesive tape strips (D-Square D101; 1.54 cm$^2$; Clinical & Derm). Each strip was applied to the skin under controlled pressure using a disc pressure instrument (D-Square D500; 255 g/cm$^2$; Clinical & Derm) for 5-10 seconds, then gently removed with tweezers and stored individually in labeled sampling vials. In patients with AD, SC sampling was performed at a clinically defined lesional site on the volar forearm (approximately 10 cm below the elbow crease) and at a contralateral, anatomically matched non-lesional site. Healthy controls were sampled at corresponding volar forearm sites.

At each sampling site, five consecutive tape strips were collected. The first two strips were discarded to minimize potential surface contamination. The third strip was designated for RNA analysis (Shima et al., 2022), the fourth strip for surface topography imaging using HS-DAFM, and the fifth strip for quantification of natural moisturizing factors (NMFs) (Kezic et al., 2009). Tape strips allocated for HS-DAFM topographic imaging were stored

at room temperature, whereas all remaining strips were immediately stored at $-80°$C until analysis.

## 2.2. Image acquisition and dataset preparation

SCN images were acquired using HS-DAFM equipped with an aluminium-coated silicon nitride probe (spring constant 30 pN/nm, CSC38/Al; MikroMasch) with a nominal tip radius of 8 nm. All measurements were performed in contact mode at constant height, with the contact force maintained below 10 nN to ensure consistent image quality and minimize sample deformation. For each SC sample, ten randomly selected regions were imaged to characterize corneocyte surface topography. Each SCN image was acquired at a resolution of $512 \times 512$ pixels, covering an area of $20 \times 20$ $\mu m^2$. All images were visually inspected by trained experts, and scans with visible artifacts (e.g., excessive noise, motion blur, or probe defects) were excluded.

Preprocessing was applied to mitigate common AFM imaging challenges, including low contrast and striping artifacts, while enhancing CNO visibility without introducing bias. Raw images first underwent Gaussian smoothing ($\sigma$=10 pixels) to reduce high-frequency noise (Gedraite and Hadad, 2011). Row-wise mean subtraction was then performed to correct horizontal striping caused by scanner drift (Canale et al., 2011). Image intensities were normalized to the [0, 1] range across all samples to ensure a consistent dynamic range. Finally, percentile-based local contrast enhancement was applied using disk-shaped morphological elements (diameters 9 and 15 pixels) as percentile filters, systematically scanning each image to amplify subtle features such as CNOs while suppressing background variation (Soille, 2004; Mukhopadhyay and Chanda, 2000; Kimori, 2011).

The resulting SCN dataset comprised 2,100 images, distributed across four clinical groups (600 from mild AD, 600 from moderate AD, 600 from severe AD, and 300 from healthy controls) and equally split between the Taiwan and Denmark cohorts (1,050 images each). This balanced multicenter design provided a heterogeneous yet well-controlled dataset for robust evaluation of SCN-derived biomarkers across disease severities and study sites.

## 2.3. Two-stage deep learning pipeline for CNO detection and segmentation

To obtain instance-level segmentations of CNOs, we implemented a two-stage deep learning pipeline that integrates real-time object detection with promptable segmentation (Figure 1). First, candidate CNOs were localized using YOLOv12, an attention-centric one-stage detector that achieves state-of-the-art accuracy-latency trade-offs by combining area attention with residual efficient layer aggregation networks (R-ELAN). Second, the resulting bounding boxes were used as box prompts for SAM3, a foundation model for promptable image and video segmentation that supports point-, box-, and mask-based queries, thereby generating instance-level CNO masks.

### 2.3.1. YOLO-based CNO detection

For CNO detection, we trained YOLOv12 models on a previously curated dataset of 300 SCN images with expert-annotated CNO bounding boxes, yielding an average of approximately 250 annotations per image and more than 74,000 labeled instances in total. The

dataset was randomly partitioned into a training set (90%) and a held-out test set (10%). Within the training set, we employed 10-fold cross-validation to obtain robust performance estimates across YOLOv12 variants and to select the model used for subsequent instance-level segmentation. In each fold, the training images were expanded three-fold using data augmentation, including adjustments to brightness ($-25\%$ to $25\%$), exposure ($-15\%$ to $15\%$), blur (up to 1 pixel), noise (up to 2% of pixels), and Mosaic augmentation, following the protocol described in previous work (Wang et al., 2024).

All YOLOv12 variants (N, S, M, L, X) were fine-tuned from checkpoints pretrained on the MS COCO dataset (Lin et al., 2014). Each variant was trained for 600 epochs on the SCN training set using stochastic gradient descent (SGD) with momentum (see Appendix A for detailed hyperparameter settings). Model complexity was quantified by the number of parameters (in millions, M) and floating-point operations (FLOPs, in gigaflops, G), and detection performance on the held-out test set was evaluated using precision, recall, AP@50, and AP@50–95. The variant achieving the highest AP@50–95 was subsequently selected to generate CNO detection bounding boxes for the multicenter AD dataset. All models were trained and evaluated on a single NVIDIA GeForce RTX 3090 GPU.

### 2.3.2. SAM-BASED INSTANCE-LEVEL SEGMENTATION

For instance-level segmentation of CNOs, we employed SAM-based models with YOLOv12 detections provided as box prompts. Specifically, we evaluated all SAM2 variants (SAM2.1-Tiny, SAM2.1-Small, SAM2.1-Base, and SAM2.1-Large), SAM3, and Cellpose-SAM, a widely used foundation model for biological image segmentation included as an off-the-shelf baseline.

To quantitatively benchmark these segmentation approaches, we curated a held-out test subset comprising 10 SCN images with expert-annotated instance masks ($\approx$2,700 labeled CNO instances in total). Model performance was evaluated using object-level metrics (precision, recall, and F1 score at mask IoU $\geq 0.5$) and pixel-level metrics. We assessed segmentation accuracy using the Dice Similarity Coefficient (DSC) for spatial overlap, and the Average Symmetric Surface Distance (ASSD) and the 95th percentile Hausdorff Distance (HD95) to quantify boundary error. No additional morphological operations (e.g., opening or closing) were applied, as image-level quality control had already excluded scans with severe artifacts (Maier-Hein et al., 2024).

### 2.4. Morphometric feature extraction and sample-level profiling

Instance-wise morphometric descriptors were computed from predicted CNO masks using region-based morphological analysis. For each CNO instance, we quantified area, perimeter, eccentricity, solidity (ratio of object area to convex hull area), major and minor axis lengths of the best-fitting ellipse, orientation, and centroid coordinates. From these primary measurements, we derived an aspect ratio as a measure of elongation, defined as

$$\text{aspect ratio} = \frac{\text{major axis length}}{\text{minor axis length}} \tag{1}$$

and a compactness index ("circularity") defined as

$$\text{circularity} = \frac{4\pi\,\text{area}}{\text{perimeter}^2} \tag{2}$$

which approaches 1 for a perfect circle. Instances with degenerate geometry (e.g., zero minor axis length or zero perimeter) that yielded undefined shape descriptors were excluded from further analysis.

For subsequent morphometric analysis, we characterized individual CNOs using six geometric descriptors: area, perimeter, eccentricity, solidity, aspect ratio, and circularity. CNO masks were first filtered by area, excluding instances below the $5^{th}$ percentile or above the $95^{th}$ percentile of the global area distribution to suppress spurious detections. For each SC tape-strip sample, the remaining per-instance features from all CNOs across the ten fields of view (FOVs) were then pooled and aggregated using the per-feature median to obtain a sample-level morphometric profile. To assess robustness of sample-level profiling to the choice of aggregation statistic, we additionally report mean-aggregated morphometric results in Appendix B. The Wilcoxon rank-sum test (Wilcoxon, 1945) was used to assess differences between independent sample groups. Samples with missing data were excluded from the analysis.

## 3. Results

### 3.1. Detection performance of YOLOv12 variants on SCN images

Across all YOLOv12 variants, CNO detection on SCN images achieved consistently high performance, with AP@50 exceeding 80% for every model (Table 1). Increasing model complexity from YOLOv12-N (2.6M parameters, 6.3G FLOPs) to YOLOv12-L (26.4M parameters, 88.5G FLOPs) led to a gradual increase in AP@50–95 from 38.37% to 41.01%, indicating improved localization performance under stricter IoU thresholds. Detection performance saturated for the largest variant, with YOLOv12-X incurring substantially higher computational cost without further improvement in AP@50–95. Considering this accuracy-efficiency trade-off, we selected YOLOv12-L as the default detector for generating CNO bounding boxes in the subsequent SAM-based segmentation and morphometric analyses.

Table 1: Comparison of detection performance and model complexity for YOLOv12 variants on the held-out test set (n = 30 SCN images). Precision, recall, AP@50, and AP@50–95 are reported as mean ± standard deviation over 10-fold cross-validation.

| Model | Parameters (M) | FLOPs (G) | Precision ↑ (%) | Recall ↑ (%) | AP@50 ↑ (%) | AP@50-95 ↑ (%) |
|---|---|---|---|---|---|---|
| YOLOv12-N | 2.6 | 6.3 | 78.32 ± 0.56 | 78.04 ± 0.57 | 81.99 ± 0.28 | 38.37 ± 0.35 |
| YOLOv12-S | 9.3 | 21.2 | 79.12 ± 0.42 | 78.70 ± 0.41 | 82.77 ± 0.23 | 39.93 ± 0.25 |
| YOLOv12-M | 20.2 | 67.1 | 79.60 ± 0.43 | 78.10 ± 0.43 | 82.79 ± 0.49 | 40.46 ± 0.33 |
| YOLOv12-L | 26.4 | 88.5 | 79.77 ± 0.47 | 78.26 ± 0.52 | 82.90 ± 0.49 | 41.01 ± 0.32 |
| YOLOv12-X | 59.1 | 198.5 | 79.73 ± 0.29 | 77.51 ± 0.68 | 82.23 ± 0.44 | 40.70 ± 0.28 |

Table 2: Comparison of segmentation performance and model complexity for Cellpose-SAM, SAM2 variants, and SAM3 on the held-out test subset (n = 10 SCN images). Models were evaluated using object-level metrics (mask IoU $\geq$ 0.5) alongside pixel-level overlap and boundary metrics.

| Model | Parameters (M) | Object-Level Metrics | | | Pixel-Level Metrics | | |
|---|---|---|---|---|---|---|---|
| | | Precision ↑ (%) | Recall ↑ (%) | F1 ↑ (%) | DSC ↑ (%) | ASSD ↓ (pixels) | HD95 ↓ (pixels) |
| Cellpose-SAM | - | 41.34 | 40.50 | 40.91 | 82.34 | 0.77 | 1.91 |
| SAM2.1-Tiny | 38.9 | 78.82 | 75.55 | 77.15 | 81.32 | 0.87 | 1.96 |
| SAM2.1-Small | 46.0 | 80.29 | 76.96 | 78.59 | 81.86 | 0.84 | 1.90 |
| SAM2.1-Base | 80.8 | 78.36 | 75.11 | 76.70 | 81.53 | 0.86 | 1.96 |
| SAM2.1-Large | 224.4 | 76.85 | 73.67 | 75.23 | 81.02 | 0.88 | 2.04 |
| SAM3 | 84.8 | 82.22 | 79.33 | 80.75 | 83.07 | 0.76 | 1.75 |

## 3.2. Segmentation performance of SAM-based models

### 3.2.1. Pixel-level benchmark of CNO instance segmentation

Using YOLOv12-L detections as box prompts, we evaluated the CNO segmentation performance of SAM2 variants and SAM3 on the held-out test subset (Table 2), and included the off-the-shelf Cellpose-SAM baseline for comparison. Among the promptable models, SAM3 achieved the best overall performance, attaining the highest object-level F1 score (80.75%) and pixel-level DSC (83.07%). It also yielded the most accurate boundary delineation, with the lowest boundary errors (ASSD = 0.76 pixels; HD95 = 1.75 pixels). The SAM2 variants showed comparable performance across both object- and pixel-level metrics. Notably, SAM2.1-Small was the best-performing SAM2 variant (F1 = 78.59%; DSC = 81.86%), while performance declined marginally with larger architectures.

In contrast, Cellpose-SAM exhibited substantially lower object-level performance (F1 = 40.91%), indicating limited ability to distinguish individual instances. Nevertheless, its pixel-level overlap remained competitive (DSC = 82.34%), with boundary errors comparable to the top-performing models (ASSD = 0.77 pixels; HD95 = 1.91 pixels). These quantitative results are consistent with the qualitative examples in Figure 2, in which Cellpose–SAM exhibits frequent false detections and notable underdetection of CNOs, whereas SAM-based models generate contours that more closely align with the ground-truth annotations.

### 3.2.2. Effect of prompt quality on SAM3 segmentation performance

To assess SAM3's sensitivity to prompt quality, we generated box prompts using YOLOv12 detectors of varying scales (N, S, M, L, X) and evaluated the resulting segmentation performance on the held-out test subset (Table 3). Overall, segmentation accuracy exhibited a positive correlation with detector capacity, with the YOLOv12-L configuration achieving the highest performance across both object- and pixel-level metrics. However, increased detection accuracy resulted in only marginal reductions in boundary error, as ASSD remained within a narrow range (0.76–0.81 pixels) despite object-level F1 scores varying from 75.33% to 80.75%. This indicates that while robust prompts are crucial for accurate object local-

ization, the intrinsic boundary delineation capability of SAM3 remains largely invariant to minor variations in prompt precision.

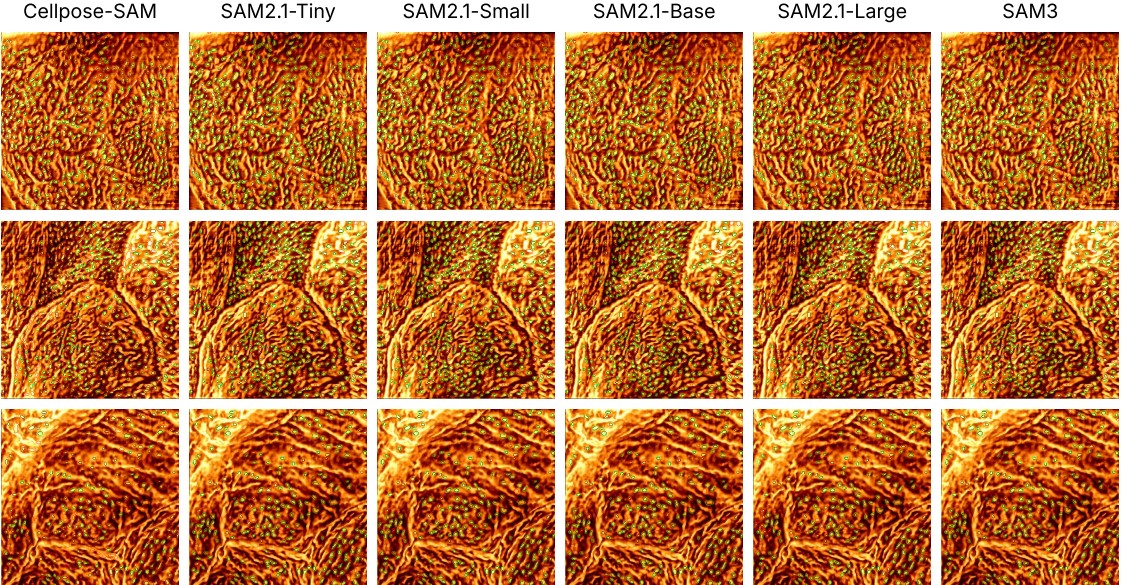

Figure 2: Qualitative comparison of CNO segmentation by Cellpose-SAM, SAM2 variants, and SAM3. Representative SCN images are shown with model predictions overlaid on ground-truth masks (black contours). Green contours denote true-positive segmentations (mask IoU ≥ 0.5), whereas red contours indicate false-positive predictions (mask IoU < 0.5 or unmatched detections).

Table 3: Evaluation of SAM3 segmentation performance using box prompts generated by YOLOv12 detectors of varying scales (N, S, M, L, X). Performance is reported using object-level metrics (Mask IoU ≥ 0.5) alongside pixel-level overlap and boundary metrics.

| Model (+SAM3) | Object-Level Metrics | | | Pixel-Level Metrics | | |
|---|---|---|---|---|---|---|
| | Precision ↑ (%) | Recall ↑ (%) | F1 ↑ (%) | DSC ↑ (%) | ASSD ↓ (pixels) | HD95 ↓ (pixels) |
| YOLOv12-N | 70.35 | 81.07 | 75.33 | 82.06 | 0.81 | 1.88 |
| YOLOv12-S | 73.84 | 79.14 | 76.40 | 82.06 | 0.81 | 1.87 |
| YOLOv12-M | 77.32 | 76.04 | 76.67 | 82.30 | 0.80 | 1.84 |
| YOLOv12-L | 82.22 | 79.33 | 80.75 | 83.07 | 0.76 | 1.75 |
| YOLOv12-X | 79.00 | 75.26 | 77.08 | 82.63 | 0.78 | 1.83 |

### 3.3. SCN morphometric analysis of healthy, non-lesional AD, and lesional AD skin

We assessed the association between SCN-derived morphometric profiles and clinical disease status by comparing three sample types within each center: healthy control skin, clinically non-lesional skin from patients with AD, and lesional AD skin. For this analysis, AD samples were pooled across mild, moderate, and severe disease (Figure 3).

In both the Taiwanese and Danish AD cohorts, the shape-related descriptors (eccentricity, solidity, circularity, and aspect ratio) exhibited consistent and statistically significant differences across the three sample types. Eccentricity and aspect ratio increased from healthy control skin to non-lesional AD skin and were highest in lesional AD skin, indicating that CNOs became progressively more elongated with increasing disease involvement. Conversely, solidity and circularity decreased along the same gradient, reflecting a shift from compact, nearly circular CNOs in healthy skin toward less compact, more irregular structures in diseased skin. Differences in area and perimeter were modest and did not reach statistical significance across sample types.

Collectively, these center-specific analyses demonstrate that segmentation-based SCN morphometric profiles capture consistent, directional alterations in CNO geometry associated with disease involvement across geographically and ethnically distinct cohorts. Elongation and loss of compactness of CNOs are already detectable in clinically non-lesional AD skin and become further accentuated in lesional sites, in line with the progressive gradients observed in the established count-based metric, the Effective Corneocyte Topographical Index (ECTI) (Wang et al., 2024).

### 3.4. SCN morphometric gradients across AD severity at clinically non-lesional sites

We evaluated whether SCN-derived morphometric profiles capture graded differences in disease severity at clinically non-lesional sites. Within each center, four clinical severity groups were compared: healthy controls and patients with mild, moderate, or severe AD (Figure 4).

In the Taiwanese cohort, the shape-related descriptors (eccentricity, solidity, circularity, and aspect ratio) exhibited statistically significant differences between healthy controls and AD severity groups, with most metrics demonstrating graded shifts across mild, moderate, and severe disease. Consistently, the Danish cohort showed a similar directional pattern, characterized by progressive elongation and reduced compactness of CNOs with increasing AD severity, albeit with smaller effect sizes and fewer statistically significant differences between healthy controls and the individual AD severity groups.

Across centers, the circularity of CNOs emerged as the most consistent discriminator, exhibiting significant differences between healthy controls and AD severity groups in both cohorts, whereas area and perimeter showed no clear association with disease severity. Notably, severe AD samples tended to contain a higher number of detected CNOs, which may have contributed to the increased variability observed in their morphometric profiles. These results indicate that segmentation-based SCN morphometric profiling aligns closely with the established count-based ECTI metric across two geographically distinct centers.

(a) Taiwanese Atopic Dermatitis Cohort

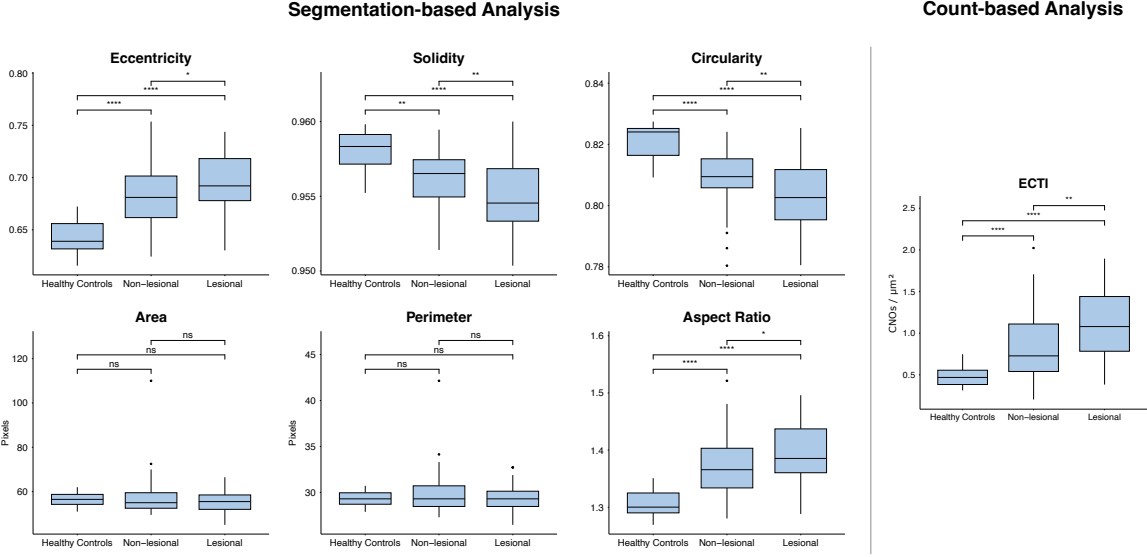

(b) Danish Atopic Dermatitis Cohort

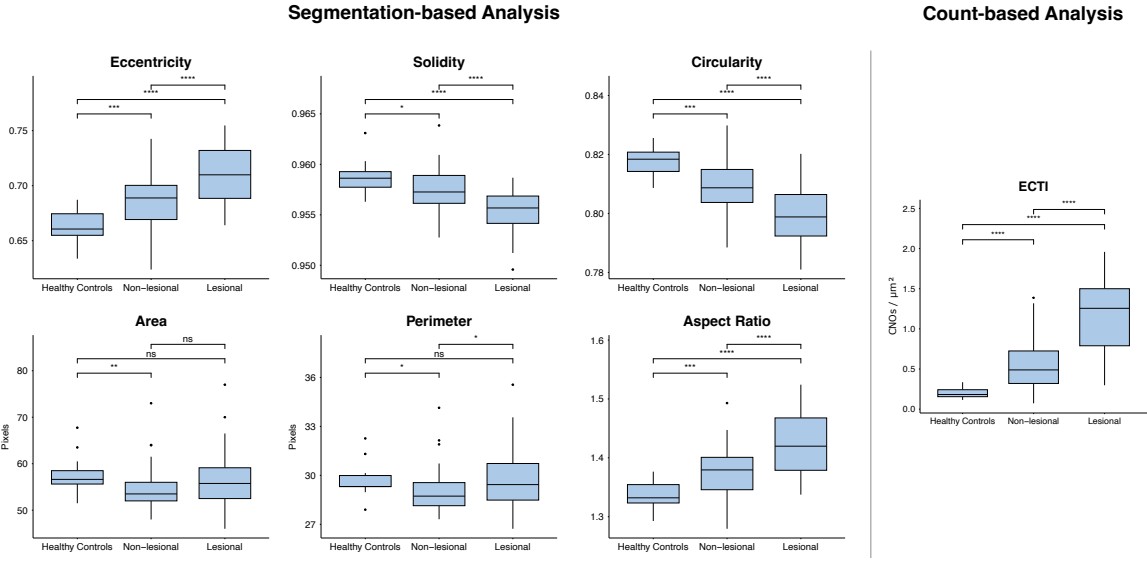

Figure 3: Center-specific analysis of SCN morphometry across healthy control skin, clinically non-lesional AD skin, and lesional AD skin in (a) Taiwanese and (b) Danish cohorts. For each cohort, the segmentation-based analysis (left) presents sample-level boxplots of CNO eccentricity, solidity, circularity, area, perimeter, and aspect ratio derived from SAM3-based instance segmentations. The count-based analysis (right) shows boxplots of the Effective Corneocyte Topographical Index (ECTI). Boxplot notation: *p < 0.05, **p < 0.01, ***p < 0.001, ****p < 0.0001; ns, not significant.

(a) Taiwanese Atopic Dermatitis Cohort

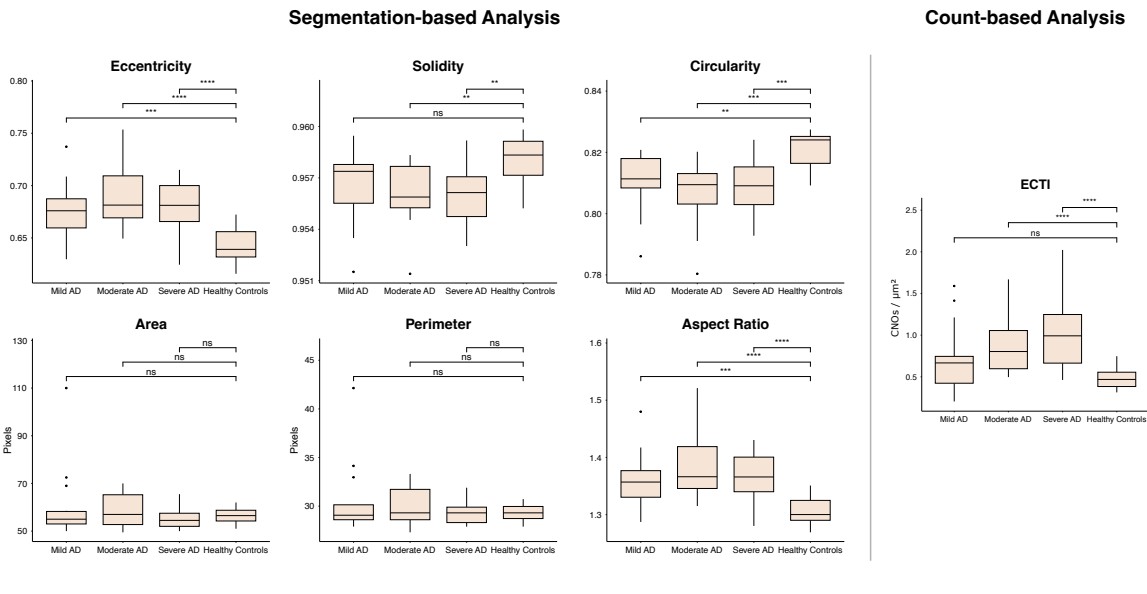

(b) Danish Atopic Dermatitis Cohort

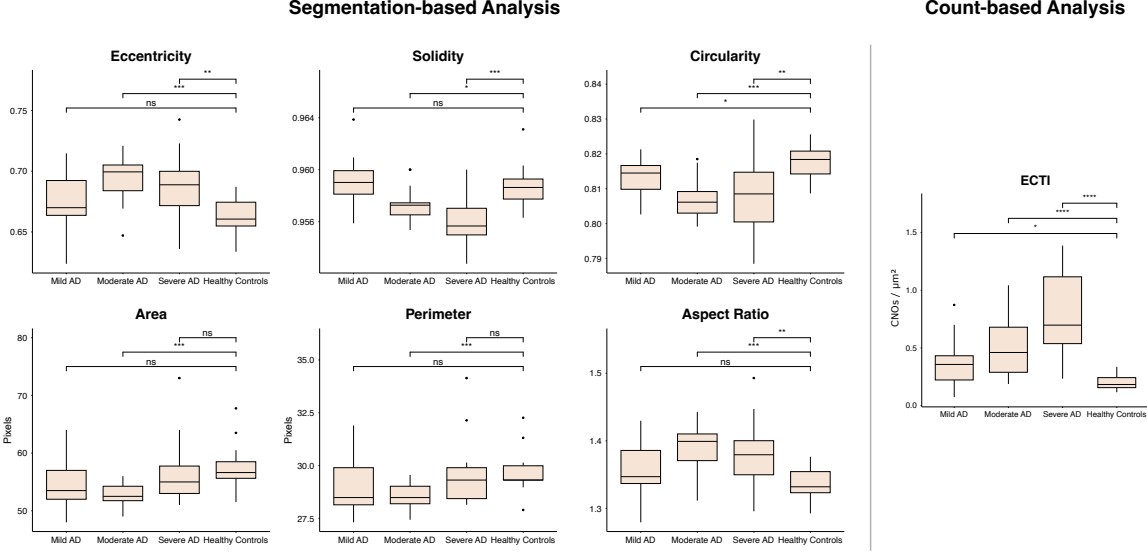

Figure 4: Center-specific analysis of SCN morphometry at clinically non-lesional sites in healthy controls and patients with mild, moderate, or severe AD in (a) Taiwanese and (b) Danish cohorts. For each cohort, the segmentation-based analysis (left) presents sample-level boxplots of CNO eccentricity, solidity, circularity, area, perimeter, and aspect ratio derived from SAM3-based instance segmentations. The count-based analysis (right) shows boxplots of the Effective Corneocyte Topographical Index (ECTI). Boxplot notation: *p < 0.05, **p < 0.01, ***p < 0.001, ****p < 0.0001; ns, not significant.

### 3.5. SCN morphometry across Fitzpatrick skin phototypes

We evaluated the influence of skin phototype on SCN morphometry by examining sample-level morphometric profiles stratified by Fitzpatrick type across healthy controls, clinically non-lesional AD skin, and lesional AD skin within each center (Figure 5). For the stratified analyses, we restricted the Taiwanese cohort to phototypes III–V and the Danish cohort to phototypes II–IV (Table 4). Phototype II in Taiwan and phototypes I and V in Denmark were either absent or represented by only 1–2 participants per clinical group, thereby precluding reliable estimation of phototype-specific differences.

Across the retained phototypes, both cohorts exhibited similar disease-related gradients in most morphometric descriptors. Eccentricity and aspect ratio increased from healthy controls to clinically non-lesional and lesional AD skin, whereas solidity and circularity decreased. Changes in area and perimeter were minimal. For a given sample type, mean morphometric values were comparable across Fitzpatrick types, and no consistent monotonic trends with increasing phototype were observed. These findings indicate that, within the represented phototype range, SCN morphometry is driven primarily by clinical disease status rather than skin phototype.

Table 4: Distribution of Fitzpatrick skin phototypes (Types I–V) among healthy controls and patients with AD in the Taiwanese and Danish cohorts. Values indicate the number of participants per center, clinical group, and skin phototype.

| Center / Clinical group | Fitzpatrick skin phototype | | | | | Total |
|---|---|---|---|---|---|---|
| | Type I | Type II | Type III | Type IV | Type V | |
| *Taiwan* | | | | | | |
| Healthy controls | 0 | 2 | 4 | 9 | 0 | 15 |
| Patients with AD | 0 | 2 | 12 | 15 | 16 | 45 |
| *Denmark* | | | | | | |
| Healthy controls | 1 | 5 | 5 | 4 | 0 | 15 |
| Patients with AD | 1 | 16 | 16 | 11 | 1 | 45 |

(a) Taiwanese Atopic Dermatitis Cohort

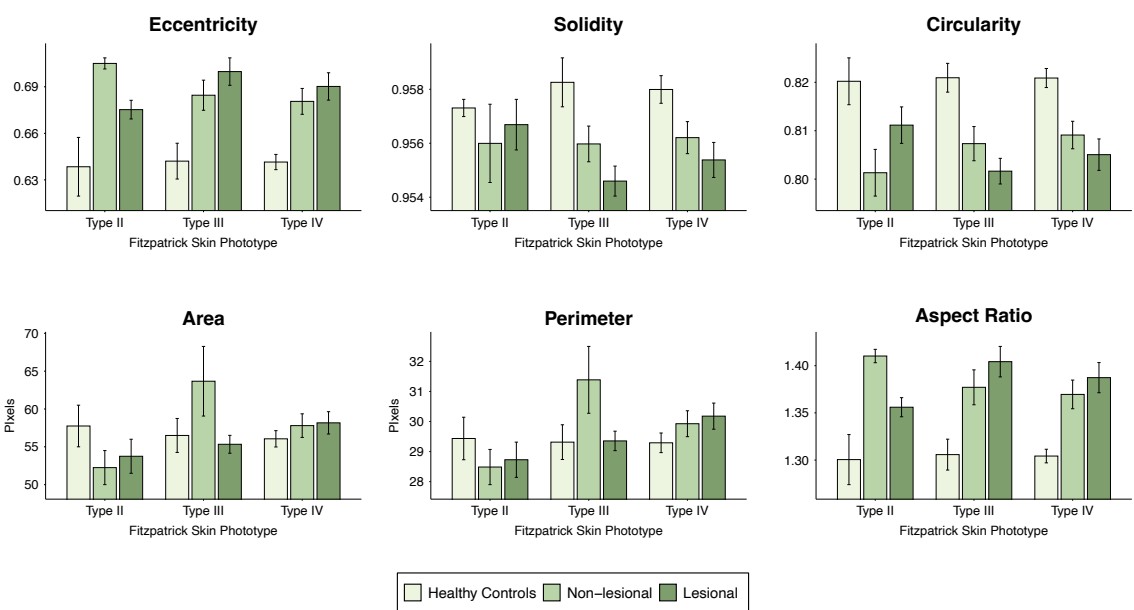

(b) Danish Atopic Dermatitis Cohort

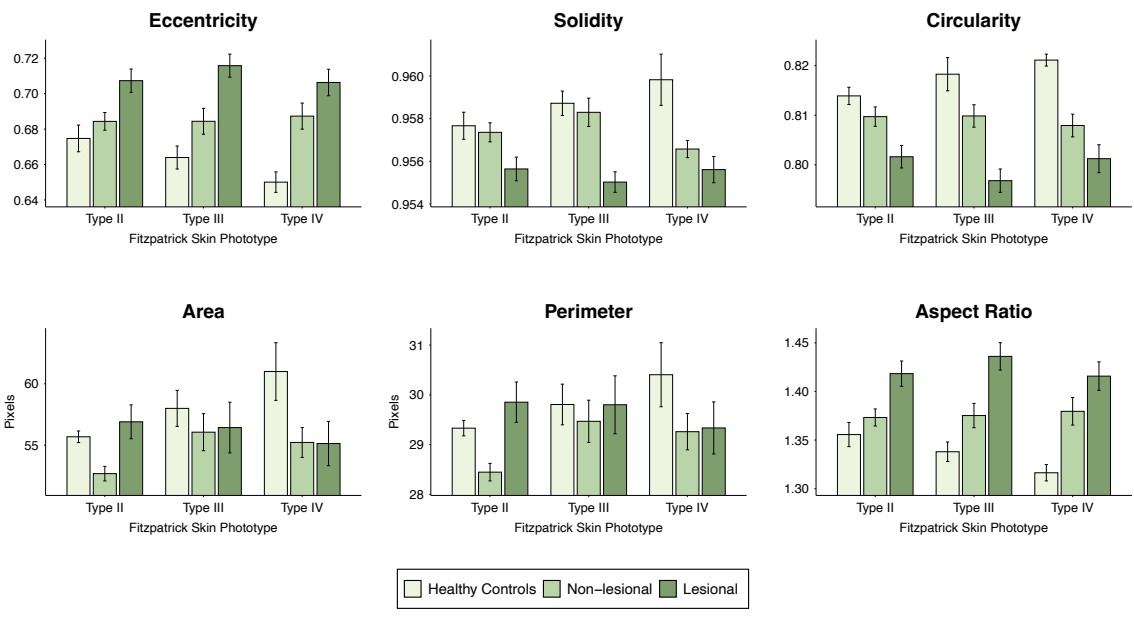

Figure 5: SCN morphometry stratified by Fitzpatrick skin phototype in (a) Taiwanese and (b) Danish cohorts. For each cohort, bar plots show sample-level mean values of CNO eccentricity, solidity, circularity, area, perimeter, and aspect ratio for healthy controls, clinically non-lesional AD skin, and lesional AD skin across the retained Fitzpatrick types (III–V for Taiwan; II–IV for Denmark). Bars indicate group means and error bars denote standard errors.

## 4. Discussion

In this multicenter study, we demonstrate that segmentation-based SCN analysis provides robust, clinically meaningful morphometric descriptors of CNO geometry that are reproducible across centers and disease severities and appear largely independent of Fitzpatrick skin phototype within the represented range. Building on prior work that quantified CNO density using count-based metrics, we show that instance-level segmentation with SAM3 enables the extraction of complementary morphometric features that capture subtle alterations in corneocyte surface architecture associated with AD. Across the Taiwanese and Danish cohorts, shape-related descriptors (eccentricity, solidity, circularity, and aspect ratio) exhibited consistent gradients from healthy control skin to non-lesional and lesional AD skin, and from mild to severe disease at clinically non-lesional sites, whereas size-related metrics (area and perimeter) contributed limited discriminative value. Overall, the findings indicate that SCN morphometry can serve as an objective, quantitative biomarker of skin barrier impairment.

The observed pattern of progressive CNO elongation and loss of compactness from healthy to clinically non-lesional and lesional AD skin indicates that segmentation-based SCN morphometry is sensitive to subtle disease-related alterations that may not be apparent on routine clinical examination. The finding that SCN morphometric profiles in non-lesional AD skin are intermediate between those of healthy and lesional AD skin supports the presence of subclinical barrier abnormalities at clinically normal-appearing skin. These gradients were consistent across centers and closely aligned with the established ECTI metric, indicating that SCN morphometry provides complementary information to count-based measures and enhances the characterization of barrier involvement across the AD severity spectrum.

Methodologically, the proposed YOLOv12-SAM3 pipeline advances SCN analysis beyond heuristic, count-based approaches by establishing a fully automated, instance-level segmentation framework based on modern deep learning. YOLOv12-L achieved a high AP@50 of approximately 82.9% for CNO detection, providing accurate and computationally efficient bounding boxes as prompts for SAM3. This two-stage pipeline outperformed the off-the-shelf Cellpose-SAM baseline on the pixel-level segmentation benchmark, achieving a DSC of 83.07% with sub-pixel boundary accuracy (ASSD = 0.76 pixels). The subsequent extraction of region-based morphometric descriptors transforms SCN from a single scalar measure (e.g., CNO counts or ECTI) into a multivariate profile that can be analyzed at both the instance and sample level, enabling more nuanced characterization of barrier-related alterations and facilitating integration into downstream statistical and machine learning models.

Several limitations should be acknowledged. First, all participants were adults recruited from two medical centers, and sampling was restricted to a single skin site, which limits generalizability to pediatric populations, other anatomical regions, and additional ethnic groups. Second, the proposed segmentation pipeline was trained and evaluated on a curated set of expert-annotated images and relied on manual quality control to exclude scans with severe artifacts. Although this reflects current best practice in SCN image analysis, translation into routine clinical workflows will require further refinement, including automated image quality assessment, reduced reliance on manual annotation, and standardized image

acquisition and analysis protocols. Third, this study focused on limited set of handcrafted features and sample-level aggregation. More sophisticated modeling of SCN morphometric distributions, spatial organization, and multiscale texture may further enhance clinical utility and robustness of SCN-derived biomarkers for specific clinical endpoints. Future work may benefit from longitudinal studies integrating SCN morphometrics, clinical severity scores, and biochemical markers (e.g., NMF levels) to determine whether CNO geometry can serve not only as a cross-sectional biomarker of current barrier impairment but also as a predictor of disease trajectory and therapeutic response.

## 5. Conclusion

This multicenter study demonstrates that segmentation-based analysis of SCN provides a robust and scalable framework for quantitative skin barrier assessment in AD, complementing established count-based metrics such as ECTI. By integrating the YOLOv12-SAM3 segmentation pipeline with downstream morphometric profiling, we show that geometric descriptors of CNOs sensitively capture subclinical barrier impairment at clinically non-lesional sites across two geographically and ethnically distinct centers, while remaining functionally independent of Fitzpatrick skin phototype within the examined range. These findings support SCN morphometry as a non-invasive, objective biomarker for quantitative skin barrier assessment and establish a methodological foundation for future applications in longitudinal monitoring, treatment response assessment, and extension to other inflammatory or barrier-related skin diseases.

## Acknowledgments

This work was supported by the LEO Foundation (LF-OC-20-000370 and LF-OC-24-001760), the Novo Nordisk Foundation (NNF22OC0076607), the National Science and Technology Council of Taiwan (NSTC 112-2314-B-002-074-MY3), and the Chan Zuckerberg Initiative grant (DAF2021-225261, https://doi.org/10.37921/644085ggkbos), an advised fund of Silicon Valley Community Foundation (funder https://doi.org/10.13039/100014989). We thank all study participants and collaborating clinical staff for their contributions to this work.

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

## Appendix A.  YOLOv12 training and augmentation hyperparameters

Table I summarizes the training configuration, loss weights, and data augmentation settings used for the YOLOv12-N/S/M/L/X models in our CNO detection pipeline, following the hyperparameter configuration described in the YOLOv12 paper (Tian et al., 2025). We adopt an SGD optimizer for 600 epochs with momentum 0.937, weight decay $5 \times 10^{-4}$, and a learning rate that decays linearly from $10^{-2}$ to $10^{-4}$. The loss terms are weighted according to the default YOLOv12 configuration, and data augmentation is implemented using the Albumentations library (Buslaev et al., 2020) with Mosaic, Mixup, copy-paste, and additional color and geometric jitter.

Table I: Training configuration and data augmentation hyperparameters for the YOLOv12-N/S/M/L/X models used for CNO detection in SCN images.

| Hyperparameters | YOLOv12-N/S/M/L/X |
|---|:---:|
| *Training Configuration* | |
| Epochs | 600 |
| Optimizer | SGD |
| Momentum | 0.937 |
| Batch size | 16 |
| Weight decay | $5 \times 10^{-4}$ |
| Warm-up epochs | 3 |
| Warm-up momentum | 0.8 |
| Warm-up bias learning rate | 0.0 |
| Initial learning rate | $10^{-2}$ |
| Final learning rate | $10^{-4}$ |
| Learning rate schedule | Linear decay |
| *Loss Parameters* | |
| Box loss gain | 7.5 |
| Class loss gain | 0.5 |
| DFL loss gain | 1.5 |
| *Augmentation Parameters* | |
| HSV saturation augmentation | 0.7 |
| HSV value augmentation | 0.4 |
| HSV hue augmentation | 0.015 |
| Translation augmentation | 0.1 |
| Scale augmentation | 0.5/0.9/0.9/0.9/0.9 |
| Mosaic augmentation | 1.0 |
| Mixup augmentation | 0.0/0.05/0.15/0.15/0.2 |
| Copy-paste augmentation | 0.1/0.15/0.4/0.5/0.6 |
| Close mosaic epochs | 10 |

## Appendix B. Mean-based morphometric analysis

To verify that the observed morphometric trends are robust and not artifacts of specific statistical descriptors, we performed a sensitivity analysis by employing mean-based aggregation for sample-level profiling. The alternative approach reproduced the same directional differences and statistical significance patterns as the median-based analysis across both cohorts, indicating that the study's conclusions are insensitive to the choice of aggregation method. Figure I and Figure II present the mean-aggregated morphometric results. Figure I compares disease status across healthy control skin, clinically non-lesional AD skin, and lesional AD skin, while Figure II shows severity-stratified comparisons at clinically non-lesional sites.

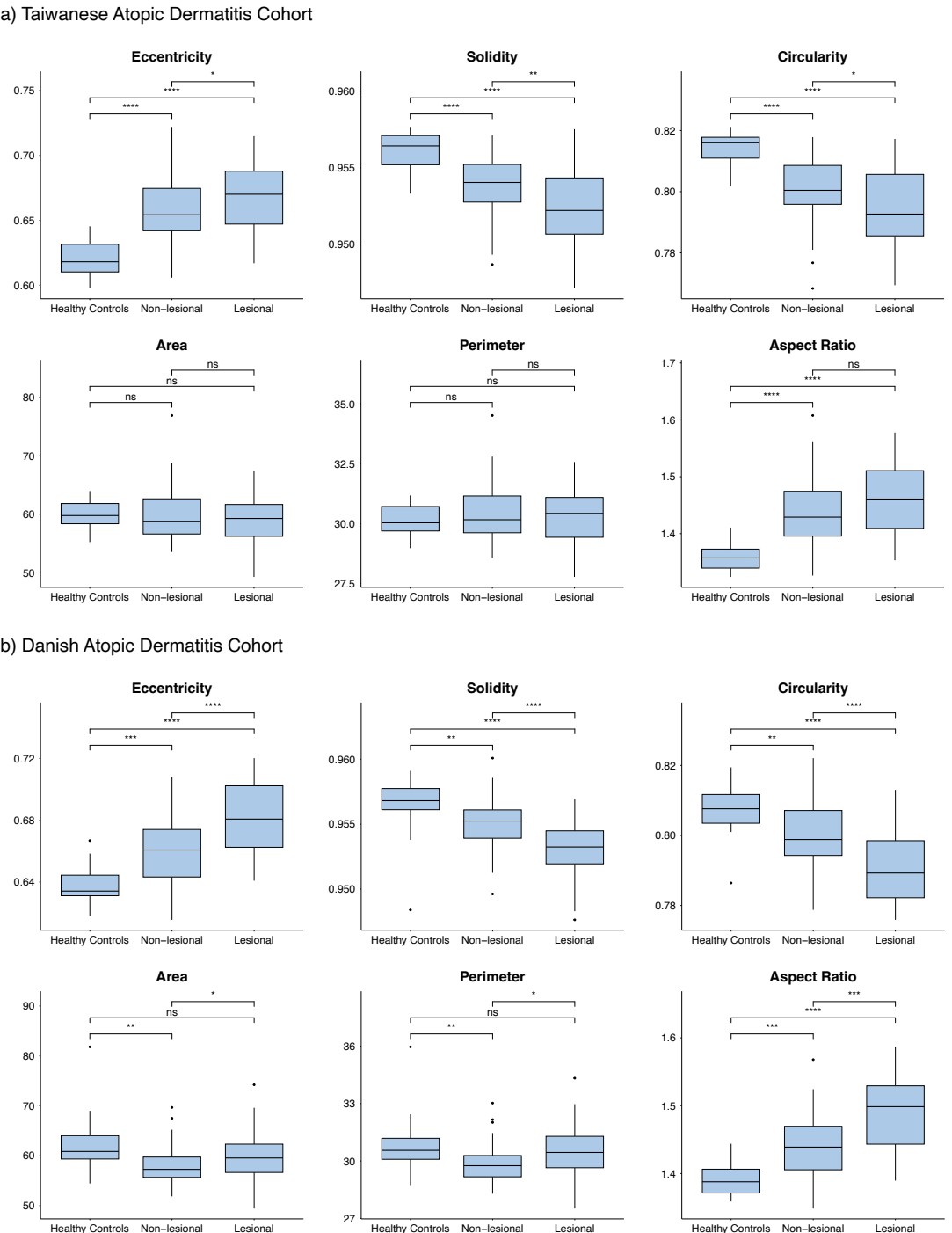

Figure I: Mean-based aggregation of SCN morphometry by disease status across healthy control skin, clinically non-lesional AD skin, and lesional AD skin in (a) Taiwanese and (b) Danish cohorts. Boxplot notation: *p < 0.05, **p < 0.01, ***p < 0.001, ****p < 0.0001; ns, not significant.

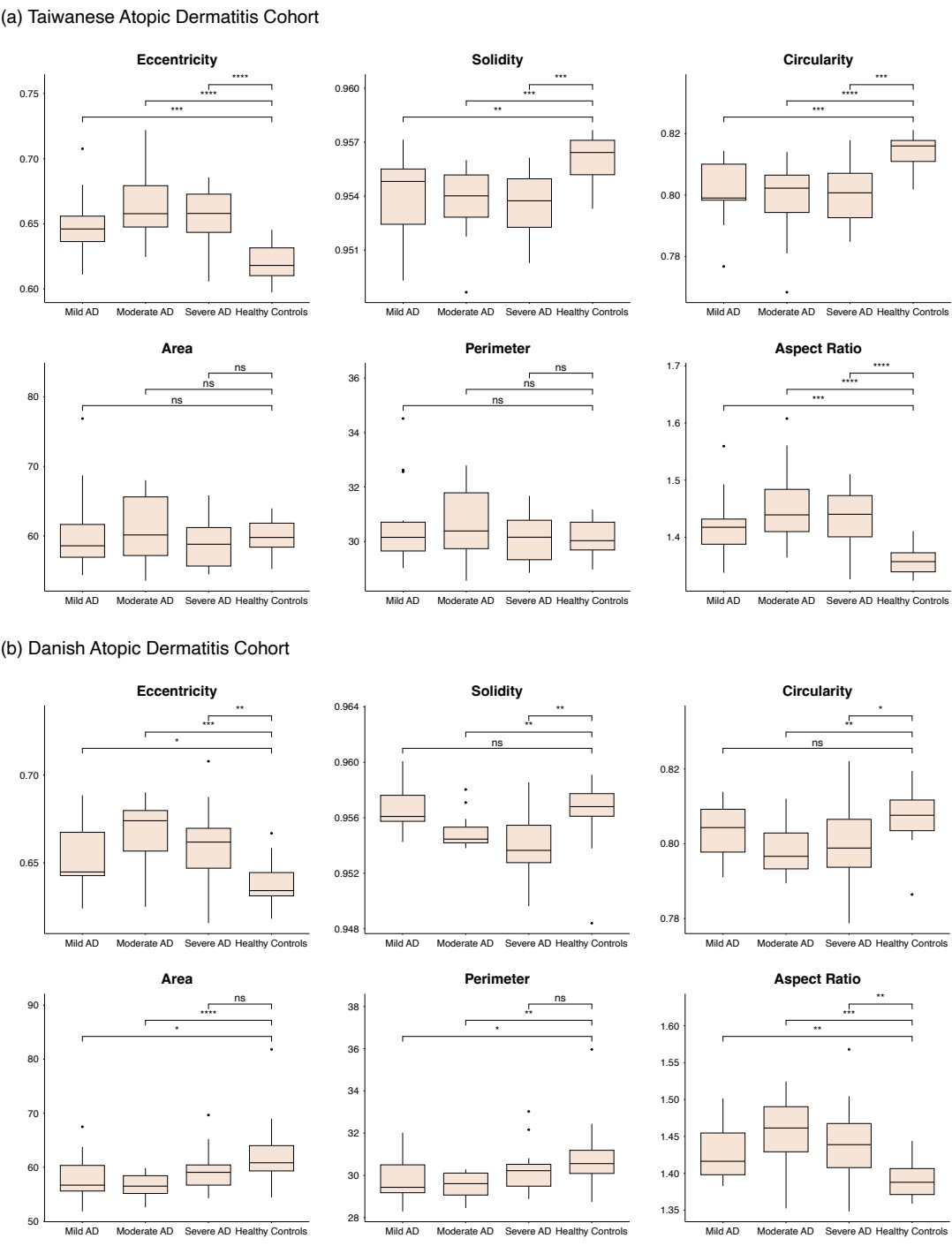

Figure II: Mean-based aggregation of SCN morphometry at clinically non-lesional sites in healthy controls and patients with mild, moderate, or severe AD in (a) Taiwanese and (b) Danish cohorts. Boxplot notation: *p < 0.05, **p < 0.01, ***p < 0.001, ****p < 0.0001; ns, not significant.

