# OpenReview forum: "Multicenter Morphometric Analysis of Stratum Corneum Nanotexture for Skin Barrier Assessment"
_MIDL.io/2026/Validation_Papers — MIDL 2026 - Validation Papers Poster_

### Official Review · Reviewer_GiTU · 2026-01-04

**Confidence:** 3
**Preliminary Rating:** 3
**Final Rating:** 4

**Summary:**

This multicenter study proposes a YOLOv12-SAM3 pipeline for morphometric analysis of stratum corneum nanotexture as an atopic dermatitis biomarker. The authors analyzed atomic force microscopy images from 90 AD patients and 30 healthy controls across Taiwan and Denmark, extracting instance-level morphometric features from circular nano-size objects. Shape descriptors showed progressive changes from healthy to non-lesional to lesional AD skin, with consistent gradients across severity levels and centers. The findings demonstrate that segmentation-based SCN morphometry complements count-based metrics and remains independent of Fitzpatrick skin phototype.

**Strengths:**

1. The balanced recruitment of 30 participants per severity group across Taiwan and Denmark provides robust cross-cultural validation. Even stratification across EASI severity levels enables comprehensive disease spectrum evaluation.
2. The YOLOv12-SAM3 pipeline achieves superior segmentation performance (AP@50: 82.9%) compared to baseline methods. Instance-level segmentation enables the extraction of six morphometric descriptors beyond simple CNO counts.
3. Morphometric features show consistent disease-related gradients across independent cohorts. The finding that non-lesional AD skin exhibits intermediate profiles suggests subclinical barrier impairment detection.
4. Detailed hyperparameter documentation and GitHub code release facilitate community adoption. Transparent preprocessing pipeline and augmentation strategies support reproducibility.

**Weaknesses:**

1. Ground truth consists only of bounding boxes, not pixel-level masks, creating a validation gap. The low IoU threshold (0.30) and box-level evaluation obscure true boundary segmentation accuracy. Perimeter-dependent metrics like circularity may be unreliable without pixel-level validation.
2. No inter-rater reliability, test-retest stability, or field-of-view variance is reported. Aggregation via median across 10 FOVs discards valuable variance information. For a biomarker intended to improve upon EASI reliability, measurement stability is essential but unaddressed.
3. Six global shape descriptors ignore spatial organization patterns potentially important in AD. No texture co-occurrence, clustering metrics, or size distribution modeling is explored. Median distribution information that may capture disease heterogeneity.
4. NMF and RNA data were collected but not analyzed alongside morphometrics.

**Detailed Comments:**

1. Figures 3-5 have small fonts
2. Paper reorganization for improved readability

**Justification Of Final Rating:**

Thanks for all the detailed reply and revision. The rebuttal has addressed all of my concerns. Therefore, I increased the rating to weak accept and do think this paper has its own contirbution to the field.

**Justification Of The Preliminary Rating:**

This paper presents sound work with contributions to SCN analysis for atopic dermatitis assessment. The finding that morphometric profiles remain independent of Fitzpatrick skin phototype strengthens generalizability claims. With revisions addressing these concerns, the reviewer is willing to change scores during the discussion session.

**Questions To Address In The Rebuttal:**

1. Can you provide Dice coefficients or boundary F1 scores on a mask-annotated validation subset? What is accuracy at stricter IoU thresholds (0.5, 0.7)?
2. Is sampling reproducible? If the same site were tape-stripped twice (consecutive strips or different days), would morphometric profiles remain consistent?
3. What is the relationship with gene expression? Can morphometrics predict NMF deficiency? Could SCN imaging substitute for biochemical assays, or do they provide orthogonal information requiring combined assessment?

---

### Official Review · Reviewer_b1ky · 2026-01-09

**Confidence:** 4
**Preliminary Rating:** 3
**Final Rating:** 4

**Summary:**

This paper presents a multicenter validation study of segmentation-based morphometric analysis of stratum corneum nanotexture (SCN) as an objective biomarker for skin barrier impairment in atopic dermatitis (AD). Using a heterogeneous dataset of over 2,000 AFM-derived SCN images from two international clinical centers, the authors propose a two-stage deep learning pipeline that combines YOLOv12 for CNO detection and SAM3 for instance-level segmentation. From the resulting segmentations, detailed morphometric descriptors of circular nano-size objects (CNOs) are extracted and aggregated at the sample level. The study demonstrates that shape-related morphometric features correlate consistently with disease presence and severity across centers and appear largely independent of Fitzpatrick skin phototype, supporting their potential clinical utility as robust, quantitative biomarkers.

**Strengths:**

1. The two-stage YOLOv12 + SAM3 design is technically reasonable, well-motivated, and empirically justified through systematic comparisons across model variants.
2.The morphometric descriptors (e.g., circularity, eccentricity) show consistent gradients across disease states, including non-lesional skin, supporting biological plausibility.
3. Inclusion of cohorts from Taiwan and Denmark with balanced disease severity enhances external validity and translational relevance.

**Weaknesses:**

1. Segmentation performance is assessed via bounding-box–derived metrics due to lack of pixel-level ground truth, which weakens confidence in fine-grained morphometric accuracy.
2. Errors from YOLO detection likely propagate to SAM-based segmentation, but sensitivity analyses on detection quality are not presented.
3. The analysis relies mainly on univariate comparisons and median aggregation, without multivariate modeling or effect size reporting to better quantify clinical relevance.

**Detailed Comments:**

1. Provide additional justification for the chosen IoU threshold (0.3) and discuss how sensitive conclusions are to this choice.
2. Expand discussion on how segmentation errors might bias specific morphometric descriptors differently (e.g., circularity vs. area).
3. Clarify how robust the morphometric findings are to alternative aggregation strategies (e.g., mean, trimmed mean, distributional descriptors).

**Justification Of Final Rating:**

The authors have effectively addressed the main technical concerns by adding a rigorous pixel-level segmentation benchmark with expert annotations and boundary-based metrics, substantially improving confidence in the reliability of the extracted morphometric features. The prompt-sensitivity analysis further demonstrates robustness of the SAM-based segmentation to variations in YOLO detection quality, alleviating concerns about error propagation, and the added aggregation analysis supports the stability of the reported trends. Remaining limitations, which should be clearly acknowledged in the final manuscript, include the reliance on primarily univariate analyses without effect size reporting or multivariate modeling, the largely qualitative discussion of descriptor-specific sensitivity to segmentation errors, and the absence of more distribution-level or predictive clinical modeling. Overall, the revisions elevate the work to an acceptable level of technical rigor, supporting a weak accept while leaving room for further strengthening.

**Justification Of The Preliminary Rating:**

The paper demonstrates solid technical execution, strong multicenter validation, and clinically meaningful findings that align well with the goals of the Validation Track. However, the indirect evaluation of segmentation quality introduces uncertainty regarding the reliability of the extracted morphometric features. While these limitations do not invalidate the conclusions, they reduce the overall impact and leave open questions about robustness and generalizability.

**Questions To Address In The Rebuttal:**

1. How sensitive are the reported morphometric trends to the quality of YOLO detections (e.g., using a smaller or noisier detector)?
2. Can the authors provide evidence (quantitative or qualitative) that segmentation masks are sufficiently accurate for reliable shape analysis, beyond box-level metrics?

---

### Official Review · Reviewer_Z3jp · 2026-01-09

**Confidence:** 4
**Preliminary Rating:** 4
**Final Rating:** 4

**Summary:**

Different from prior SCN work that mainly relies on counting CNOs, this paper proposes a two-stage pipeline that detects CNOs using a fine-tuned YOLOv12 detector and then segments instances using SAM3. This enables extraction of instance-level CNO morphometric features for sample-level profiling. The authors evaluate detection and segmentation performance and assess whether the resulting morphometric profiles show consistent differences across healthy, non-lesional AD, and lesional AD skin, as well as across EASI-defined severity strata.

**Strengths:**

1. The use of a large, multicentre biopsy cohort is valuable and much closer to real diagnostic workflows, which makes the evaluation more clinically relevant. The multicenter design with balanced severity groups improves external validity compared with prior single-center SCN studies.
2. The motivation and method description are well written and easy to follow, and the implementation details are helpful for reproducibility.
3. The paper compares multiple YOLOv12 variants and multiple segmentation backbones, and includes qualitative visualizations that help interpret typical failure cases.
4. The morphometric analysis shows consistent associations between SCN signals and disease status as well as EASI severity strata, strengthening the translational framing.

**Weaknesses:**

1. SAM3 is a strong foundation segmentation model, but it’s not described as being trained on SCN or medical-imaging data. Using it off-the-shelf without domain adaptation or fine-tuning may limit segmentation quality and introduce systematic errors, especially for tiny objects and unusual texture distributions.
2. The segmentation evaluation is indirect. It relies on box-level ground truth rather than pixel masks. This makes it hard to judge true boundary fidelity and whether the extracted morphometrics are accurate at the instance level.
3. Clinical utility remains unclear. The paper shows group differences, but does not directly evaluate patient-level tasks, which limits the evidence for real-world decision support.

**Detailed Comments:**

1. Add a small pixel-mask benchmark. Even a modest subset with pixel-level masks would allow reporting Dice/IoU and boundary metrics to validate that SAM3 produces morphometrically faithful shapes.
2. Explicit cross-center generalization tests.
3. Add a patient-level task. A simple baseline that predicts severity group or EASI score from morphometric profiles would greatly strengthen translational relevance and show whether morphometrics add value beyond count-based metrics.

**Justification Of Final Rating:**

The rebuttal addressed my main technical concern. However, clinical utility remains to be determined. The current evidence is largely group-level, and the paper still does not provide patient-level performance so the real-world decision impact is still undecided.

**Justification Of The Preliminary Rating:**

The paper targets an objective, non-invasive biomarker for AD barrier impairment, and the multicenter dataset with balanced severity groups is a big plus. The pipeline is also clearly described, with thorough model comparisons. I’m still a bit uncertain about the segmentation validity and I’d also like to see a simple patient-level task and a more explicit cross-center generalization test to better support the translational claim.

**Questions To Address In The Rebuttal:**

1. Do you have any evidence that the SAM3 masks are morphometrically accurate?
2. Why no SAM fine-tuning adaptation?
3. Can you demonstrate patient-level utility and compare it to simpler count-based metrics?

---

### Author Rebuttal · Authors · 2026-01-24

**Rebuttal:**

(The revised manuscript has been uploaded with the changes highlighted in yellow.)

We appreciate the reviewers’ constructive feedback, which has significantly improved our manuscript. In this revision, we have strengthened the validation of our pipeline by adding:

* **Pixel-level segmentation benchmark:** We curated a held-out test subset of 10 SCN images with expert-annotated pixel masks (≈2,700 labeled CNO instances) and report Dice Similarity Coefficient (DSC) and boundary metrics (ASSD, HD95) to directly validate mask fidelity for morphometric analysis (Section 3.2.1, Table 2).

* **Sensitivity to prompt quality:** We assessed SAM3 robustness using box prompts from YOLOv12 variants of different scales (N, S, M, L, X), quantifying how detector variability propagates to segmentation quality and morphometric fidelity (Section 3.2.2, Table 3).

* **Alternative aggregation strategies:** To assess robustness to the choice of aggregation statistic, we additionally report mean-aggregated morphometric analyses alongside our primary median-based results (Appendix B).
​​
* **Minor Revisions:** We reorganized the manuscript to improve readability, and increased the font size in Figures 3–5 to enhance visual legibility and clarity.

Further details are provided in the individual responses below.

**Supporting Material:**

/attachment/3a7b7ec91aa6bb030855d67b3dc245099cec3810.pdf

---

### Meta-Review · Area_Chair_XBbA · 2026-02-09

**Recommendation:** Accept (Poster)
**Confidence:** 4

**Metareview:**

While this validation paper had originally received rather critical or mixed reviews - the authors have successfully improved the quality through the revision and rebuttal phase. All reviewers now recommend (weak) acceptance and see a valuable contribution in the work. There are some remaining issues/limitations highlighted by one reviewer, which should be clearly acknowledged in the final manuscript: "the reliance on primarily univariate analyses without effect size reporting or multivariate modeling, the largely qualitative discussion of descriptor-specific sensitivity to segmentation errors, and the absence of more distribution-level or predictive clinical modeling"
Besides this I recommend acceptance as poster.

---

### Decision · Program_Chairs · 2026-02-14

Accept (Poster)